# Association of Circulating Neutrophils with Relative Volume of Lipid-Rich Necrotic Core of Coronary Plaques in Stable Patients: A Substudy of SMARTool European Project

**DOI:** 10.3390/life13020428

**Published:** 2023-02-02

**Authors:** Silverio Sbrana, Antonella Cecchettini, Luca Bastiani, Annamaria Mazzone, Federico Vozzi, Chiara Caselli, Danilo Neglia, Alberto Clemente, Arthur J. H. A. Scholte, Oberdan Parodi, Gualtiero Pelosi, Silvia Rocchiccioli

**Affiliations:** 1CNR Institute of Clinical Physiology, 54100 Massa, Italy; 2Department of Clinical and Experimental Medicine, University of Pisa, 56126 Pisa, Italy; 3CNR Institute of Clinical Physiology, 56124 Pisa, Italy; 4Fondazione Toscana Gabriele Monasterio, 54100 Massa, Italy; 5Fondazione Toscana Gabriele Monasterio, 56124 Pisa, Italy; 6Department of Cardiology, Leiden University Medical Center, 2333 ZA Leiden, The Netherlands

**Keywords:** coronary artery disease, neutrophils, platelets, cytokines, flow cytometry, coronary CT angiography, lipid-rich necrotic core volume

## Abstract

Background and Aims: Coronary atherosclerosis is a chronic non-resolving inflammatory process wherein the interaction of innate immune cells and platelets plays a major role. Circulating neutrophils, in particular, adhere to the activated endothelium and migrate into the vascular wall, promoting monocyte recruitment and influencing plaque phenotype and stability at all stages of its evolution. We aimed to evaluate, by flow cytometry, if blood neutrophil number and phenotype—including their phenotypic relationships with platelets, monocytes and lymphocytes—have an association with lipid-rich necrotic core volume (LRNCV), a generic index of coronary plaque vulnerability, in a group of stable patients with chronic coronary syndrome (CCS). Methods: In 55 patients, (68.53 ± 1.07 years of age, mean ± SEM; 71% male), the total LRNCV in each subject was assessed by a quantitative analysis of all coronary plaques detected by computed tomography coronary angiography (CTCA) and was normalized to the total plaque volume. The expression of CD14, CD16, CD18, CD11b, HLA-DR, CD163, CCR2, CCR5, CX3CR1, CXCR4 and CD41a cell surface markers was quantified by flow cytometry. Adhesion molecules, cytokines and chemokines, as well as MMP9 plasma levels, were measured by ELISA. Results: On a per-patient basis, LRNCV values were positively associated, by a multiple regression analysis, with the neutrophil count (*n°*/µL) (*p* = 0.02), neutrophil/lymphocyte ratio (*p* = 0.007), neutrophil/platelet ratio (*p* = 0.01), neutrophil RFI CD11b expression (*p* = 0.02) and neutrophil–platelet adhesion index (*p* = 0.01). Significantly positive multiple regression associations of LRNCV values with phenotypic ratios between neutrophil RFI CD11b expression and several lymphocyte and monocyte surface markers were also observed. In the bivariate correlation analysis, a significantly positive association was found between RFI values of neutrophil–CD41a+ complexes and neutrophil RFI CD11b expression (*p* < 0.0001). Conclusions: These preliminary findings suggest that a sustained increase in circulating neutrophils, together with the up-regulation of the integrin/activation membrane neutrophil marker CD11b may contribute, through the progressive intra-plaque accumulation of necrotic/apoptotic cells exceeding the efferocytosis/anti-inflammatory capacity of infiltrating macrophages and lymphocytes, to the relative enlargement of the lipid-rich necrotic core volume of coronary plaques in stable CAD patients, thus increasing their individual risk of acute complication.

## 1. Introduction

Atherosclerotic coronary artery disease (CAD) is a chronic non-resolving inflammatory disease of the vessel wall, involving several cells of the innate immune system. In our recent studies, we evaluated blood monocytes’ phenotypic aspects associated with differently polarized systemic inflammatory environments in a group of stable patients under an optimal treatment for chronic coronary syndrome (CCS). We found that CAD severity, as assessed by the degree of maximal vessel stenosis, was associated with a prevalent systemic pro-inflammatory polarization, whilst, on the other hand, the relative amount of dense-calcium plaque deposition was associated with an anti-inflammatory type of blood monocyte polarization [1,2].

Recent studies provide strong evidence of the presence of neutrophils in atherosclerotic plaques and of their primary location in inflamed areas [3,4,5]. In particular, it has been demonstrated that they promote atherosclerotic plaque instability, either by contributing to matrix degradation through the release of proteases, or as a source of not cleared apoptotic bodies and necrotic cells into the so-called necrotic core, a typical feature of advanced complicated atherosclerotic lesions, in which their accumulation overwhelms the phagocytic capacity of macrophages [6]. In agreement with these findings, the number of neutrophils in the arterial intima has been also found to positively correlate with the signs of plaque vulnerability in humans and mice [7] as well as neutrophil myeloperoxidase (MPO) inhibition to reduce the necrotic core in the experimental atherosclerosis of the aortic roots in mice [8].

Up to now, most of the studies concerning the role of neutrophils in the evolution of atherosclerotic lesions have been based on the immunohistochemical phenotypic characterization of the cell types present within the vascular wall. Only a few studies have considered the association between the neutrophil and lymphocyte numerical ratios in peripheral blood (NLR), taken as an index of systemic inflammation, with the extent of atherosclerotic diseases and with the volumetric quantification of plaque components [9,10].

In this study of 55 subjects participating in a SMARTool clinical trial and with coronary plaques detected by computed tomography coronary angiography (CTCA) [1,2], we aimed to evaluate, for the first time, the cross-sectional association between circulating neutrophil number and their phenotypic features—assessed by flow cytometry—and the total plaque lipid-rich necrotic core volume (LRNCV) obtained after normalization for the global plaque volume. This study is expected to provide further insights, also as possible therapeutic targets, into the understanding of pathophysiological mechanisms that can contribute to plaque instability and to the onset of acute ischemic events in stable CAD patients.

## 2. Materials and Methods

### 2.1. Patients

The evaluation has been restricted to a subgroup of patients (*n*° = 55), recruited as part of the European Project SMArtool (Clinical-Trials.gov Identifier: NCT04448691), in which the availability of a CTCA assessment of coronary plaque composition was fully matched with flow cytometry data as well as with the clinical-demographic and immune-biochemical parameters summarized in Table 1.

### 2.2. CTCA and Quantitative Image Analysis

CTCA procedures and quantitative image analysis have previously been described in detail [2]. In brief, CTCA-assessed lipid-rich necrotic core volume (LRNCV, a.u. = arbitrary units) was quantified for each patient as the ratio of the sum of LRNCV values, expressed in mm^3^ in all detected plaques in between the Hounsfield units interval previously used [11], to the sum of total plaque volume (mm^3^).

### 2.3. Biochemical and Hematological Analyses

Differential blood cell and platelet (PLT) counts (*n*° of cell/µL) were carried out by using an automated cell analyzer. Serum lipid profile, general biochemical parameters and ELISA tests were performed as already described [1,2].

### 2.4. Flow Cytometry Analysis

Flow cytometry data were collected as previously described [9]. In brief, within 1 h after EDTA-anticoagulated blood collection, the expression levels of markers CD14, CD16, CD18, CD11b, HLA-DR, CD163, CCR2, CCR5, CX3CR1, CXCR4 and CD41a were quantified—as percentage of positivity (%+) and relative fluorescence intensity (RFI)—on the three main leukocyte populations of peripheral blood (neutrophils, monocytes, lymphocytes). Triple staining combinations of PC5-conjugated anti-human CD14 (cat. A07765) (from Beckman Coulter), FITC-conjugated anti-human CD16 (cat. 555406) (from BD Pharmingen), PE-conjugated anti-human CD11b (cat. 555388), CD163 (cat. 556018), CD41a (cat. 555467) (from BD Pharmingen), and PE-conjugated anti-human CD18 (cat. FAB1730P), HLA-DR (cat. FAB4869P), CCR2 (cat. FAB151P), CCR5 (cat. FAB1802P), CX3CR1 (cat. FAB5204P) and CXCR4 (cat. FAB173P) (from R&D) were employed [9].

The gating identification strategy and phenotypic quantification of circulating monocytes has already been described [1]. Neutrophils and lymphocytes have been gated by using a dot-plots of SSC (side scattering light) vs. CD14 and SSC vs. FSC (forward scattering light), respectively (Figure 1A,C). The CD41a antibody, recognizing the gpIIb glycoprotein of the gpIIb/IIIa surface platelet complex (CD41/CD61), was also used to quantify both the frequency of circulating heterotypic platelet–leukocyte aggregates (PLAs) and the number of platelets bound per cell (as RFI) (Figure 1B). Neutrophil and lymphocyte (Figure 1D) phenotypic quantifications, both as %+ and RFI, were carried out using an overlaid histogram subtraction analysis, as reported in our previous work [1].

### 2.5. Statistical Analysis

Continuous data are presented as mean ± mean standard error (SEM) and categorical variable is presented as number of patients and percentage. The statistical comparison between CAD severity groups was performed by ANOVA (with Bonferroni’s correction) for continuous data, either as source data or after appropriate numerical transformation of categorical variables. For the multiple linear regression analysis, the LRNCV value was considered as the dependent variable. The number of independent variables included in the present statistical model 1 adjustment (Table 2), and already partially described in our previous work [2], was expanded by the addition of six soluble parameters, concerning both the lipid metabolism (e.g., LDL cholesterol and triglycerides levels, HDL/LDL ratio, body mass index, use of statin therapy) and the neutrophil functional state (e.g., MMP9). The addition of these further six parameters was performed in order to establish an association, as far as possible free from the influence of lipid metabolism and systemic inflammation, between blood cells’ phenotypic features and the lipid-rich necrotic core relative volume within the coronary plaques. Bivariate correlations were also used to investigate the relationships existing between the following: (a) PLT count and VCAM-1 plasma levels; (b) neutrophil CD11b expression (as RFI) and circulating levels of IL-6 and IL-10/IL-6 ratio; and (c) neutrophil expression (as RFI) of pro-adhesive/chemotactic receptors and the number of platelet bound per neutrophil (complexes neutrophil–CD41a+, as RFI).

As previously noted [2], age- and sex-related differences in parameters included in the above-mentioned multiple regression analysis model were already considered in the calculation of the Framingham risk score. All statistical analyses were carried out with the Stat View 5.0 software program (SAS Institute, Cary, NC, USA). A *p* value < 0.05 was considered statistically significant.

## 3. Results

### 3.1. Patients Clinical Characteristics and Plasma Biochemistry

The most relevant clinical and immune-biochemical parameters used in the selected group of patients (*n°* = 55) for the multiple regression analysis adjustment are reported in Table 1, according to their CAD severity classes, as previously reported [12,13]. The mean LRNCV values (a.u.) are also reported in the last row of the table to investigate their possible association with CAD stenosis severity; no statistically significant differences were observed between the three groups.

The multiple regression analysis between the above parameters—excluding gender and age, which were already included in the computation of the Framingham risk score—and the LRNCV values are reported in Table 2 (model 1 adjustment). The statistical calculation of the LRNCV-associated multiple regression analysis is based on a complete per-patient-based matching between all the immune-biochemical and clinical parameters listed in Table 2.

An evident positive correlation trend, although not completely statistically significant, existed between LRNCV and IFN-γ (*p* = 0.06). Moreover, the ratio IL-10/IL-6 (51.30 ± 5.76, mean ± SEM)—not shown in the table—correlated significantly and inversely with the LRNCV values (*p* = 0.03; regression coefficient = −0.001).

### 3.2. Relationships between Leukocyte Subsets Count, Leukocyte Subsets Count Ratios, Neutrophil/PLT Ratio and LRNCV Values

The overall white blood cell (WBC) count (*n°*/µL) and the neutrophil cell count (*n°*/µL) were positively associated in the multiple regression with the LRNCV values. Only the neutrophil/WBC ratio, the neutrophil/lymphocyte ratio and the neutrophil/platelet (PLT) ratio showed positive correlations with the LRNCV values. The above observations are summarized in Table 3.

### 3.3. Relationships between Leukocyte Phenotypic Features and LRNCV Values

No significant positive association was found in the multiple regression analysis between the lymphocyte and monocyte surface markers’ expression and the LRNCV values.

On the other hand, the flow cytometry analysis of neutrophil phenotype revealed a significant positive association (*p* = 0.0201; regression coefficient = 0.001) in the multiple regression between the CD11b molecule expression, as RFI, and the LRNCV values.

Moreover, in the bivariate linear correlation, the neutrophil CD11b expression (as RFI) correlated positively with the IL-6 plasma levels (*p* = 0.0043; R = 0.379) and inversely with the IL-10/IL-6 ratio (*p* = 0.0099; R = 0.345).

Finally, the neutrophil CD11b expression (as RFI) correlated positively (*p* < 0.0001; R = 0.592) in the bivariate linear correlation with the number of platelets bound per neutrophil, expressed in terms of the RFI of the complexes neutrophil–CD41a+ (neutrophil–platelet aggregates, NPAs) identified by flow cytometry. The cumulative graphic representation of the three above-mentioned bivariate linear correlation plots is reported in Figure 2.

In our study, we observed that the neutrophil–platelet interaction also modulated the expression levels of other neutrophil receptors involved in cellular adhesion and chemotactic responses (see Appendix A).

### 3.4. Relationships between Leukocyte-Platelet Adhesion Index and LRNCV Values

As suggested in our previous work [14], the number of platelets bound per leukocyte (expressed as RFI of the leukocyte–platelet aggregates CD41a+) can be considered directly proportional to the PLT *n°*/µL and inversely proportional to the leukocyte *n°*/µL. Within this relationship, the function of proportionality is identified precisely by the *Leukocyte-PLT Adhesion Index*, a normalized parameter indicating a set of reciprocal functional interactions between the specific leukocyte subset and platelets.

On this basis, in the present study, we observed that only the *Neutrophil-PLT Adhesion Index* correlated positively and significantly (*p* = 0.0109; regression coefficient = 0.012) in the multiple regression with the LRNCV values.

### 3.5. Relationships between Leukocyte Receptor Ratios and LRNCV Values

We have observed that the receptor expression ratios significantly and positively associated (as RFI) with the LRNCV values in the multiple regression analysis were present only in the neutrophil vs. lymphocyte and neutrophil vs. monocyte groups in the phenotypic comparison. Furthermore, they only involved the neutrophil expression of CD11b (as RFI). These observations are summarized in Table 4.

## 4. Discussions

### 4.1. Study Results

In this study, the CTCA-assessed LRNCV values of coronary plaques in stable CAD patients were found to be independently associated not only with the number of circulating neutrophils—in terms of both their absolute value and relative prevalence when compared to other blood leukocyte populations and platelets (numerical ratios)—but also with the increased expression (as RFI) of the integrin molecule CD11b. The CD11b expression (Mac-1 subunit belonging to the beta2-integrin subfamily), in fact, is up-regulated following leukocyte activation, leading to a stable inter-cell contact (adhesion) with ICAM-1 located on endothelial cells [15]. Furthermore, the activation-induced up-regulation of CD11b is associated with an increased respiratory burst activity of phagocytes [16].

Additionally, the strong bivariate linear correlations, respectively, direct and inverse, observed in our study between the neutrophil expression of CD11b (as RFI) and the blood values of IL-6 and IL-10/IL-6 suggest that the increased cellular expression of this molecule is supported by a systemic immunological environment oriented towards a prevalent pro-inflammatory condition. Furthermore, according with the literature [17,18], our results confirm the important role of neutrophil-bound platelets in generating an “outside–in” mechanism of signaling able to module the neutrophil phenotype and to promote the development of coronary atherosclerotic lesions containing a wider lipid-rich necrotic core. The latter aspect, in particular, seems to be independently linked to an increased capacity of the neutrophil–platelet mutual interaction, expressed for the first time in our study in terms of the *Neutrophil–PLT adhesion index*. The above observations suggest that the calculation of this normalized index, together with the quantification of neutrophil CD11b expression, might represent a useful biomarker even in more complex immunological settings aimed at understanding the mechanisms of necrotic core development within atherosclerotic plaques.

### 4.2. Comparison with Similar Studies

The relationship between blood leukocyte and platelet populations and the different morphological components of coronary atherosclerotic plaque has been assessed so far only in terms of cellular numerical counts and ratios [7,9]. Furthermore, to date, the evaluation of the activation status of circulating neutrophils in the presence of coronary atherosclerotic disease has revealed evident phenotypic and functional cellular changes only in the presence of acute syndromes [19,20,21]. The recent demonstration of a statistically significant association between the plasma levels of MMP9, a neutrophil granule protease known to represent an amplification mechanisms of neutrophil function [18], and the adverse morphological features of coronary plaque progression—including the size of the necrotic core—in patients with CCS and stable lesions, has provided indirect evidence of the possible role played by these phagocytes in the long-term evolution of coronary atherosclerosis [22].

In this cross-sectional study, we have investigated for the first time by flow cytometry the presence of a possible statistical association between markers of the activation status of circulating neutrophils, as well as between their phenotypic functional interaction with platelets and other blood leukocyte populations, and the extent of LRNCV within coronary atherosclerotic plaques in patients under an optimal treatment for stable CAD.

Confirming and, at the same time, expanding previous observations from the literature [3,4], our data suggest that long-lasting neutrophil activation, characterized both by the up-regulation of the CD11b integrin molecule expression and by increased pro-adhesive functional interaction with circulating platelets, may lead to a sustained adhesion of phagocytes to a dysfunctional endothelium, followed by their subsequent increased migration into evolving atherosclerotic plaques.

The statistically significant associations, observed for the first time in our study, between the blood neutrophil/lymphocyte and neutrophil/monocyte phenotypic ratios and the LRNCV values seem to be in accord with those of the literature data [23], indicating that a sustained neutrophil migration into the vascular wall, when saturating the efferocytosis capacity of tissue macrophages and the anti-inflammatory properties of infiltrating lymphocytes, results in increased neutrophil necrosis, with neutrophil extracellular traps’ (NETs) release and the formation of a highly inflammatory necrotic core [24,25].

Taken together, our data highlight the importance of blood neutrophils’ phenotype and function in chronic inflammatory conditions, such as atherosclerosis, and support the observations of a recent immunological research forum indicating that these cells, thus far generally considered transcriptionally silent, are actually able to sense and adapt to subtle environmental changes, even at steady-state and homeostatic conditions, by active transcriptional plasticity [26].

### 4.3. Study Limitations

The statistical significance of the results found could be conditioned by the small number of patients studied. Furthermore, given the small number of patients studied, a gender-subgroup-based statistical evaluation was not accomplished.

Therefore, further studies on larger patient populations are mandatory to identify neutrophil surface molecules as possible biomarkers predictive of LRNC lesions’ generation and enlargement within coronary plaques in stable CAD clinical settings.

In particular, a broader flow cytometry assessment of surface markers more specific to a neutrophil activation state (e.g., CD66b, CD177) [27,28] and chemotactic response to IL-8 (e.g., CXCR1 and CXCR2) [29] may help validate the hypothesis that the volumetric buildup of LRNC lesions within advanced atherosclerotic plaque is mainly due to the long-lasting migration of activated (primed) neutrophils from the blood into the vascular wall.

Moreover, a larger flow cytometry quantification of the expression of the main counter–receptors pairs known to be involved in neutrophil–platelet heterotypic aggregation, the platelet P-selectin (CD62P) and glycoprotein (GP)Ibα (CD42b), as well as the neutrophil P-selectin glycoprotein ligand-1 (CD162) and Mac-1 complex (CD18/CD11b) [30,31,32], could provide useful insights; additionally, this could aid the identification of possible therapeutic targets, as well as clarify the role played by this important platelet-induced amplification mechanism of neutrophil activation and endothelial pro-adhesiveness in influencing atherosclerotic plaque progression.

An essential clue for our understanding of the pathophysiological mechanisms responsible for the progression of atherosclerotic disease in humans, associated with the enlargement of the plaque necrotic core, might derive also from the evaluation of the aforementioned neutrophil markers in translational animal models of experimentally induced atherosclerosis.

## 5. Conclusions

In this study, we evaluated the association between CTCA-assessed coronary plaques’ lipid-rich necrotic core normalized volumes in patients with stable CAD and in terms of the following: (i) circulating neutrophil phenotypes, (ii) blood neutrophil–platelet interactions, and (iii) blood neutrophil to leukocyte (monocytes and lymphocytes) phenotypic ratios.

The main findings of the study are as follows: (1) The number of circulating neutrophils and the expression level (RFI) of the integrin molecule CD11b were significantly associated in the multiple regression analysis with higher LRNCV values. (2) The plasma IL-6 cytokine level and the IL-10/IL-6 cytokine ratio correlated directly and inversely, respectively, with the neutrophil CD11b expression, further supporting its modulation by a systemic immunological environment mainly oriented towards a pro-inflammatory condition. (3) The neutrophil CD11b expression (RFI) correlated linearly with the number of platelets bound per neutrophil, expressed as the RFI of the complexes neutrophil–CD41a+ (neutrophil–platelet aggregates RFI, NPAs (RFI)). (4) The mutual functional interaction between neutrophils and platelets, quantified in terms of the *Neutrophil–PLT adhesion index*, was independently and positively associated with the LRNCV values. (5) The blood phenotypic ratios (as RFI) between the neutrophil integrin CD11b and a group of monocyte–lymphocyte multifunctional markers correlated independently and positively with the LRNCV values. (6) Imaging angiographic techniques, capable of providing an accurate quantitative and qualitative characterization of atherosclerotic plaques, can fundamentally support the validation of the pathophysiological mechanisms involved in the progression of coronary atherosclerotic disease.

In conclusion, an increased long-lasting activation status of circulating neutrophils, mainly evidenced by the up-regulation of the integrin/activation molecule CD11b and probably sustained by the simultaneous action of positive autocrine and paracrine functional amplification mechanisms, could determine the progressive accumulation of neutrophils within coronary atherosclerotic plaques. The subsequent pro-inflammatory necrotic death of phagocytes, overwhelming the efferocytosis and anti-inflammatory capacity of infiltrating macrophages and lymphocytes, could contribute to the development and enlargement of LRNCV in stable coronary plaques.

## Figures and Tables

**Figure 1 life-13-00428-f001:**
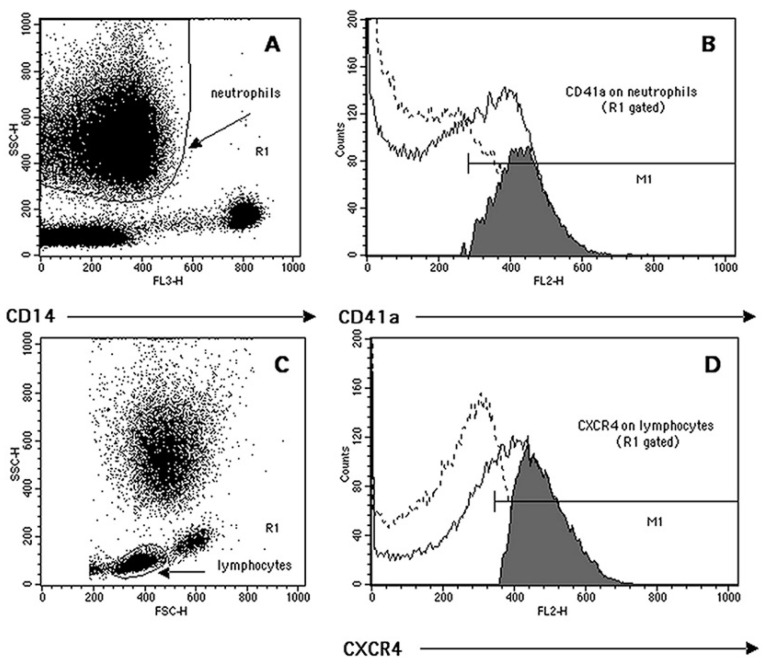
Representative example of flow cytometry quantification of complexes neutrophil–CD41a+ (NPAs, neutrophil–platelet aggregates) and lymphocyte markers’ expression. (**A**) Neutrophil cluster identification (region R1) based on its low CD14 expression (FL3) and side-scattering (SSC) morphological characteristics. (**B**) The R1-based histogram’s subtraction analysis (positive events (continuous line) minus isotype control (dotted line)) was used to quantify both the percentage of complexes CD41a+ (grey events in M1) and their RFI (median of M1 histogram minus median of the isotype control). (**C**) Lymphocyte cluster identification based on its forward- (FSC) and side-scattering characteristics (region R1). (**D**) Representative quantification of lymphocyte CXCR4 expression following the overlay histogram’s subtraction analysis (see above).

**Figure 2 life-13-00428-f002:**
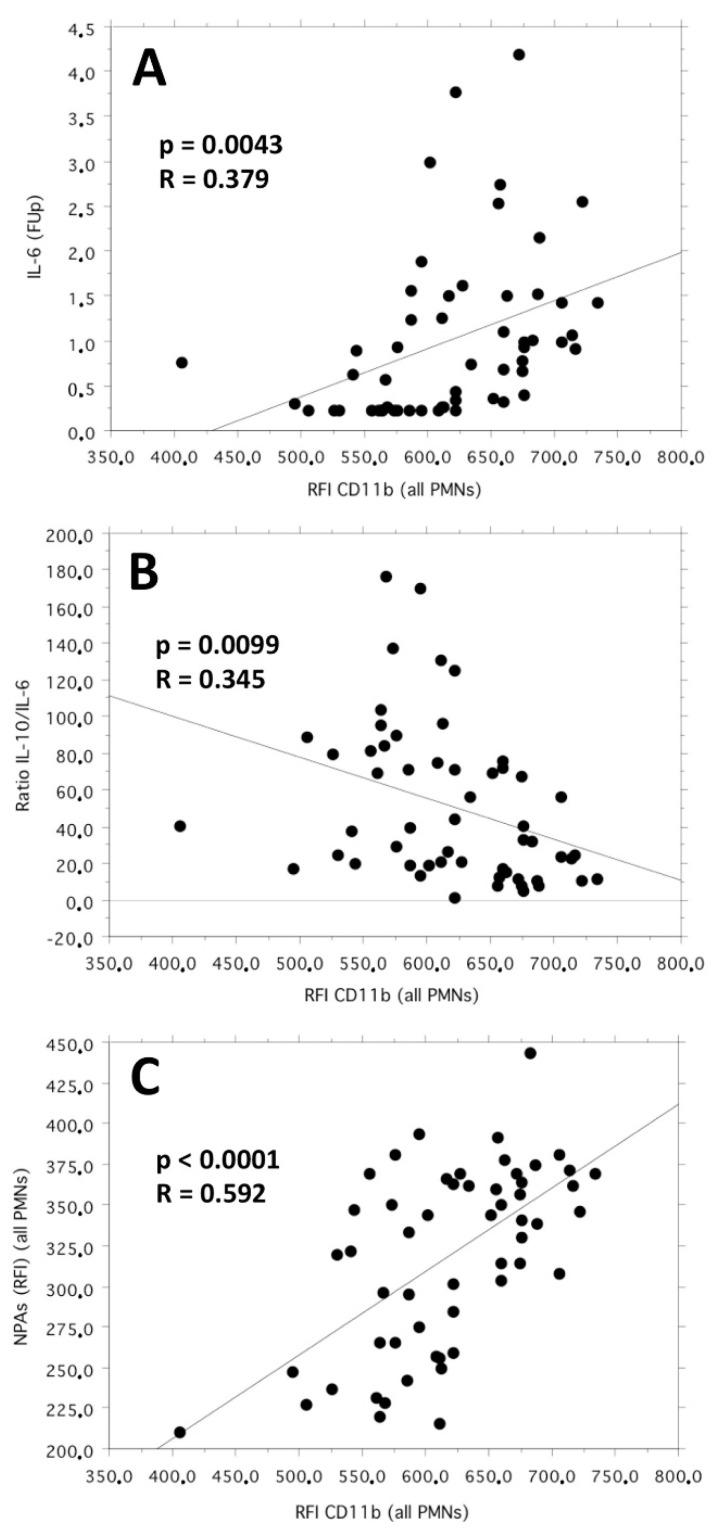
Bivariate linear correlation plots between CD11b neutrophil expression (as RFI) and the follow-up values of IL-6 plasma levels (pg/mL) (plot (**A**)), IL-10/IL-6 cytokine ratio (plot (**B**)) and neutrophil–platelet aggregates (NPAs, as RFI values of the complexes neutrophil–CD41a+) (plot (**C**)).

**Table 1 life-13-00428-t001:** Clinical and immuno-biochemical parameters, reported by CAD severity classes, in the selected group of patients (*n*° = 55).

	All Patients (*n°* = 55)	CAD1 (*n°* = 16)	CAD2 (*n°* = 21)	CAD3 (*n°* = 18)	ANOVA *p* Value
Age (years)	68.53 ± 1.07	65.18 ± 2.25	70.28 ± 1.72	69.50 ± 1.50	Ns
Gender (M/F, *n°*)	39/16	12/4	12/9	15/3	Ns
Framingham Risk Score (a.u.) (FRS)	15.40 ± 0.46	14.43 ± 1.09	16.09 ± 0.56	15.44 ± 0.75	Ns
Diabetes, *n°* (%)	17 (30.91)	2 (3.64)	6 (10.91)	9 (16.37)	0.059 *
Oral antidiabetics, *n°* (%)	15 (27.28)	2 (3.64)	5 (9.09)	8 (14.54)	Ns
Statin use, *n°* (%)	41 (74.54)	10 (18.19)	14 (25.45)	17 (30.91)	0.059 * ^
Statin dosage, (mg/die)	13.27 ± 1.49	8.75 ± 2.01	13.09 ± 2.72	17.50 ± 2.43	Ns
Creatinine (mg/dL)	0.85 ± 0.03	0.89 ± 0.05	0.78 ± 0.04	0.90 ± 0.05	Ns
Hs-CRP (mg/dL)	0.44 ± 0.10	0.53 ± 0.18	0.33 ± 0.07	0.50 ± 0.24	Ns
ICAM-1 (ng/mL)	225.53 ± 13.62	241.77 ± 24.83	222.04 ± 21.22	215.15 ± 26.03	Ns
VCAM-1 (ng/mL)	635.15 ± 22.39	719.15 ± 59.19	547.93 ± 17.10	662.24 ± 27.97	0.004 ^§^ ^
IL-6 (pg/mL)	1.01 ± 0.13	1.30 ± 0.28	0.66 ± 0.10	1.16 ± 0.24	Ns
IFN-γ (pg/mL)	32.04 ± 1.69	34.10 ± 4.52	30.52 ± 1.97	31.97 ± 2.44	Ns
IL-10 (pg/mL)	27.26 ± 1.76	40.02 ± 2.91	23.70 ± 2.51	20.07 ± 1.51	<0.0001 ^§^ *
TNF-α (pg/mL)	69.74 ± 3.11	73.33 ± 8.71	67.46 ± 4.03	69.21 ± 3.29	Ns
MCP-1 (pg/mL)	173.14 ± 8.79	184.20 ± 11.66	177.58 ± 13.98	158.14 ± 18.82	Ns
IL-8 (pg/mL)	1.97 ± 0.23	1.93 ± 0.46	1.56 ± 0.35	2.48 ± 0.41	Ns
RANTES (pg/mL)	141.25 ± 14.26	141.27 ± 24.66	144.77 ± 26.87	137.18 ± 22.47	Ns
Fractalkine (pg/mL)	0.96 ± 0.20	1.10 ± 0.35	1.32 ± 0.40	0.42 ± 0.24	Ns
MMP9 (ng/mL)	90.98 ± 11.44	130.65 ± 31.63	78.19 ± 12.57	70.63 ± 12.30	0.038 *
LDL cholesterol (mg/dL)	105.04 ± 4.61	110.19 ± 8.66	110.43 ± 8.33	94.17 ± 6.47	Ns
Triglycerides (mg/dL)	133.47 ± 6.59	128.00 ± 13.62	140.38 ± 11.99	130.28 ± 8.57	Ns
HDL/LDL Ratio	0.57 ± 0.03	0.53 ± 0.05	0.56 ± 0.04	0.61 ± 0.06	Ns
BMI (Body Mass Index)	27.26 ± 0.49	28.09 ± 0.85	26.79 ± 0.71	27.05 ± 1.00	Ns
**LRNCV (a.u.)**	0.25 ± 0.01	0.26 ± 0.02	0.25 ± 0.01	0.23 ± 0.01	Ns

Data are presented as mean ± SEM or as number (n) and percentage (%), when appropriate. The ANOVA *p* (Bonferroni post-hoc): * *CAD1/CAD3*, ^§^
*CAD1/CAD2*, ^^^
*CAD2/CAD3*; *p* < 0.05: statistically significant; Ns: not significant. Abbreviations: Hs-CRP, high-sensitivity C-reactive protein; ICAM-1, intercellular adhesion molecule-1; VCAM-1, vascular cell adhesion molecule-1; IL-6, interleukin-6; IFN-γ, interferon-gamma; IL-10, interleukin-10; TNF-α, tumor necrosis factor-alpha; MCP-1, monocyte chemoattractant protein-1; IL-8, interleukin-8; RANTES, Regulated on Activation, Normal T cell Expressed and Secreted (also CCL5, C-C motif ligand 5); Fractalkine (also C-X3-C motif ligand 1); MMP9, matrix metallopeptidase 9; LDL, low-density lipoprotein cholesterol; HDL, high-density lipoprotein cholesterol; LRNCV, lipid-rich necrotic core volume; a.u. = arbitrary units.

**Table 2 life-13-00428-t002:** Multiple regression analysis (model 1 adjustment) between clinical and immuno-biochemical parameters and LRNCV values in the selected group of patients (*n*° = 55).

	LRNCV (a.u.) (*n°* = 55)	
	Regression Coefficient	*p*-Value
Framingham risk score (a.u.)	−0.003	0.479
Diabetes	−0.015	0.811
Oral antidiabetics therapy	0.026	0.690
Statin therapy use	−0.001	0.982
Statin therapy dosage (mg/die)	−1.284 × 10^−5^	0.992
Creatinine (mg/dL)	−0.012	0.855
Hs-CRP (mg/dL)	0.001	0.970
IL-6 (pg/mL)	0.001	0.942
ICAM-1 (ng/mL)	−2.265 × 10^−4^	0.089
VCAM-1 (ng/mL)	−7.039 × 10^−5^	0.401
IFN-γ (pg/mL)	0.003	0.066
IL-10 (pg/mL)	−3.992 × 10^−4^	0.689
TNF-α (pg/mL)	−0.001	0.327
MCP-1 (pg/mL)	−1.718 × 10^−4^	0.418
IL-8 (pg/mL)	0.001	0.842
RANTES (pg/mL)	−3.252 × 10^−5^	0.768
Fractalkine (pg/mL)	0.011	0.217
MMP9 (ng/mL)	2.220 × 10^−4^	0.185
LDL cholesterol (mg/dL)	0.001	0.110
Triglycerides (mg/dL)	−1.697 × 10^−4^	0.619
HDL/LDL ratio	0.040	0.638
BMI	0.004	0.269

*p* < 0.05: statistically significant (by multiple regression analysis, model 1 adjustment); LRNCV, lipid-rich necrotic core volume; a.u. = arbitrary units; see Table 1 for other abbreviations.

**Table 3 life-13-00428-t003:** Positive associations in the multiple regression analysis between blood leukocyte subset counts (*n*°/µL), leukocyte subset count ratios, neutrophil/PLT ratio and LRNCV values in the selected group of patients (*n*° = 55).

	LRNCV (a.u.) (*n°* = 55)	
	Regression Coefficient	*p*-Value
WBC (*n°*/µL)	1.225 × 10^−5^	0.053
Neutrophil (*n°*/µL)	1.750 × 10^−5^	0.021
Neutrophil (*n°*/µL)/WBC (*n°*/µL) Ratio	0.254	0.054
Neutrophil (*n°*/µL)/Lymphocyte (*n°*/µL) Ratio (NLR)	0.026	0.007
Neutrophil (*n°*/µL)/PLT (*n°*/µL) Ratio	4.234	0.014

*p* < 0.05: statistically significant (by multiple regression analysis, model 1 adjustment); LRNCV, lipid-rich necrotic core volume; a.u. = arbitrary units. Abbreviations: WBC, white blood cells; PLT, platelet.

**Table 4 life-13-00428-t004:** Positive associations in the multiple regression analysis between leukocyte receptor phenotypic ratios (as RFI) and LRNCV values in the selected group of patients (*n*° = 55).

	LRNCV (a.u.) (*n°* = 55)	
	Regression Coefficient	*p*-Value
**Neutrophils (N) vs. Lymphocytes (L) Phenotypic Ratios (RFI)**		
CD11b (N)/CD11b (L)	0.074	0.009
CD11b (N)/CD18 (L)	0.133	0.001
CD11b (N)/CCR5 (L)	0.090	0.017
CD11b (N)/CCR2 (L)	0.071	0.011
CD11b (N)/CXCR4 (L)	0.069	0.053
**Neutrophils (N) vs. Monocytes (M) Phenotypic Ratios (RFI)**		
CD11b (N)/CCR5 (M)	0.038	0.031
CD11b (N)/CCR2 (M)	0.136	0.001
CD11b (N)/HLA-DR (M)	0.193	0.009
CD11b (N)/CD16 (M)	0.030	0.035
CD11b (N)/CD163 (M)	0.165	0.0002

*p* < 0.05: statistically significant (by multiple regression analysis, model 1 adjustment); LRNCV, lipid-rich necrotic core volume; a.u. = arbitrary units. Abbreviations: CCR5, C-C motif chemokine receptor type 5; CCR2, C-C motif chemokine receptor type 2; CXCR4, C-X-C motif chemokine receptor type 4; HLA-DR, human leukocyte antigen-DR; CD, cluster of differentiation.

## Data Availability

Not applicable.

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
