# Peer review of "Association of Circulating Neutrophils with Relative Volume of Lipid-Rich Necrotic Core of Coronary Plaques in Stable Patients: A Substudy of SMARTool European Project"

_life, 2023, doi:10.3390/life13020428_

Round 1
Reviewer 1 Report
The paper “Association of circulating neutrophils with relative volume of 2 lipid-rich necrotic core of coronary plaques in stable patients: a 3 sub-study of SMARTool clinical trial” from Sbrana et al. reports on the possible association of the vulnerability of coronary plaques with the neutrophils number/phenotype. The study is well conducted and the paper is clearly written. Minor points: the authors stated that “Serum lipid profile, general biochemical parameters 97 and ELISA tests have been performed as already described [1,2]”. In one of the cited references it has been reported that “all the biomarkers were determined in duplicates”. The authors should clearly indicate, for example in methods section, the number of replicates assays used in this work.
Author Response
In the methods section, it has been indicated that the immuno-biochemical determinations have been performed in duplicate.
Reviewer 2 Report
The article is devoted to topical issues of atherosclerosis, the study of its cellular and molecular basis. It should be noted that the strength is the translation of knowledge of fundamental cardiology into clinical cardiology. Data have been obtained on multiple associations between cellular and molecular markers, the significance of which in the progression of atherosclerosis, and their potential as markers of atherosclerosis has been convincingly shown earlier in experimental study. Unfortunately, the authors were only able to reproduce part of the experimental data, the significance of which for clinical cardiology is too early to speak. Another strength is that the identified associations convincingly show the possibilities of computed angiography in the qualitative and quantitative characterization of atherosclerotic plaques. However, the authors do not focus on this fact in the conclusion.In addition, it is necessary to present some of the associations in the form of infographics or drawings. An important point is the lack of a formulation of the hypothesis that the authors tested. In this regard, the relevance section is also very extensive. The same can be said about the conclusion, which contains many assumptions.
Author Response
A) In the Limits of the Study (paragraph 4.3 of the Discussion Section) it has been underlined the need to study experimental translational models of atherosclerotic disease in order to obtain a better understanding of cellular mechanisms responsible for plaque progression and necrotic core enlargement in humans.
B) In the Conclusions section it has been underlined the key role of imaging techniques for the validation of both clinical and experimental studies aimed to investigate the pathophysiological mechanisms involved in coronary atherosclerotic disease progression.
C) In the Results section (paragraph 3.3) a graphic representation of linear correlations existing between neutrophil CD11b expression (as RFI) and IL-6 plasma levels, IL-10/IL-6 cytokine ratio and RFI values of NPAs (Neutrophil-Platelet-Aggregates, complexes Neutrophil-CD41a+) at follow-up, is reported in Figure 2.
D) In the Conclusions section it has been also hypothesized that higher levels of CD11b expression, responsible for the chronic accumulation of circulating neutrophils within the vessel wall, are the result of different and conbined autocrine and paracrine functional amplification mechanisms.
Reviewer 3 Report
The study is well-designed and the results sufficiently presented. Authors represented a solid work, however minor corrections should be made:
1. Check English language (style, grammar) through all manuscript and make corrections such as:
· Abstract section, Line 36: “At bivariate correlation analysis...”
· Discussion section, Line 292: “Furthermore, given the small number if patients studied,...
· Materials and Methods section, Line 87: “...immune-biochemical parameters..”
2. Change the term:
· Results section, Line 158: “Clinical and biohumoral parameters, reported...”
3. There is a small number of patients included in the study. How was checked the normality distribution of the continuous variables? Are data normally distributed?
4. There is a strong recommendation to explain the abbreviations mentioned in the tables.
5. There is a strong recommendation to give three numbers after dot in p value (Tables 1 – 4).
6. There is a strong recommendation to change titles in Results section, because not all associations are significant:
· Line 221, Table 4: “Significantly positive associations, at multiple regression analysis,. ...”
· Line 183, Table 3.
Author Response
A) In the Abstract section, Discussion section and Materials and Methods section, the words “at”, “if” and “immune-biochemical”, have been replaced with words “to”, “of” and “immuno-biochemical”, respectively.
B) The term “biohumoral” has been changed with “immuno-biochemical”.
C) We agree with the reviewer about the small size of the sample studied. Indeed, as indicated in the limitations of the study, it was not possible to perform a statistical gender distinction; moreover, the need to carry out these evaluations on a higher number of patients is evident. Most of the variables used have a normal distribution; furthermore, as known from the literature, the normal distribution of the variables is not necessary in multiple regression models.
D) The abbreviations in the Tables have been explained.
E) Only three numbers after dot have been reported in the Tables statistics.
F) The titles of Tables 3 and 4 have been changed, by replacing the terms “significantly positive associations” with “positive associations”.
Reviewer 4 Report
Silverio Sbrana et al. presented in Life is undoubtedly relevant and makes a significant contribution to the understanding of atherogenesis. The material and methods are well described. However, figure 1A can be improved, the neutrophil population is slightly shifted to the left. Authors must also indicate the catalog numbers and manufacturer of the markers used (CD14, CD16, 101 CD18, CD11b, HLA-DR, CD163, CCR2, CCR5, CX3CR1, CXCR4, and CD41a) . In addition, half of the citations are older than 5 years; authors should include more recent references in the list of references.
Author Response
A) The slight shift to the left of the neutrophil cluster in Figure 1 is normal as it depends on the expression variability, in any case low, of the CD14 molecule (FL3 channel in our study) on this cell population. Therefore, the flow cytometric identification of the neutrophil cluster appears to be correct.·
B) As suggested, in the Materials and Methods section (paragraph 2.4., Flow Cytometry Analysis) the indications concerning the manufacturer and the product codes of the monoclonal antibodies used in the study have been included.·
C) In the Discussion Section (paragraph 4.2., Comparison with Similar Studies) a recent reference (n°25) (Immunology 2022) concerning the role played by the neutrophil clearance in immunity and tissue inflammation has been added.